# Near Chromosome-Level Genome Assembly and Annotation of *Rhodotorula babjevae* Strains Reveals High Intraspecific Divergence

**DOI:** 10.3390/jof8040323

**Published:** 2022-03-22

**Authors:** Giselle C. Martín-Hernández, Bettina Müller, Christian Brandt, Martin Hölzer, Adrian Viehweger, Volkmar Passoth

**Affiliations:** 1Department of Molecular Sciences, Swedish University of Agricultural Sciences, 75007 Uppsala, Sweden; giselle.martin@slu.se (G.C.M.-H.); bettina.muller@slu.se (B.M.); 2Institute for Infectious Diseases and Infection Control, Jena University Hospital, 07743 Jena, Germany; christian.brandt@med.uni-jena.de; 3Method Development and Research Infrastructure, MF1 Bioinformatics, Robert Koch Institute, 13353 Berlin, Germany; hoelzerm@rki.de; 4Institute of Medical Microbiology and Virology, University Hospital Leipzig, 04103 Leipzig, Germany; adrian.viehweger@medizin.uni-leipzig.de

**Keywords:** *Rhodotorula babjevae*, de novo hybrid assembly, nanopore sequencing, genome divergence, ploidy

## Abstract

The genus *Rhodotorula* includes basidiomycetous oleaginous yeast species. *Rhodotorula babjevae* can produce compounds of biotechnological interest such as lipids, carotenoids, and biosurfactants from low value substrates such as lignocellulose hydrolysate. High-quality genome assemblies are needed to develop genetic tools and to understand fungal evolution and genetics. Here, we combined short- and long-read sequencing to resolve the genomes of two *R. babjevae* strains, CBS 7808 (type strain) and DBVPG 8058, at chromosomal level. Both genomes are 21 Mbp in size and have a GC content of 68.2%. Allele frequency analysis indicates that both strains are tetraploid. The genomes consist of a maximum of 21 chromosomes with a size of 0.4 to 2.4 Mbp. In both assemblies, the mitochondrial genome was recovered in a single contig, that shared 97% pairwise identity. Pairwise identity between most chromosomes ranges from 82 to 87%. We also found indications for strain-specific extrachromosomal endogenous DNA. A total of 7591 and 7481 protein-coding genes were annotated in CBS 7808 and DBVPG 8058, respectively. CBS 7808 accumulated a higher number of tandem duplications than DBVPG 8058. We identified large translocation events between putative chromosomes. Genome divergence values between the two strains indicate that they may belong to different species.

## 1. Introduction

Oleaginous yeasts have received considerable attention in recent years due to many potential biotechnological applications of microbial lipids. *Rhodotorula* species are basidiomycetous oleaginous yeasts whose lipid production accounts for more than 70% of dry cell weight. They show high tolerance to inhibitors, enabling them to convert lignocellulosic hydrolysates into lipids [1,2,3,4]. Microbial lipids from *R. babjevae* and other oleaginous yeasts have a fatty acid composition similar to vegetable oils and represent an environmentally and ethically suitable alternative raw material for the production of biofuels, oleochemicals, feed, and food additives [2,5,6]. Under nitrogen-limited conditions, *R. babjevae* can simultaneously accumulate biotechnologically important enzymes, glycolipids, and carotenoids [5]. Glycolipids from *R. babjevae* have promising environmental applications in biodegrading hydrocarbon pollutants and replacing synthetic compounds and chemical surfactants [7,8,9]. They are also attractive for other applications in various industrial sectors due to their antifungal, antibacterial, antiviral, and anti-carcinogenic activities [7,8,9,10]. However, it is desirable to obtain more robust *R. babjevae* strains to overcome the high production costs of microbial lipids and biosurfactants.

There are currently no methods described for the molecular manipulation of *R. babjevae* strains. To date, several genomes from *Rhodotorula* sp. have been sequenced including different strains of *R. toruloides*, *R. graminis* WP1, and *R. glutinis* ZHK. Of these, some have only been determined using short-read sequencing technologies or lack gene annotation [3,11,12,13,14,15,16,17,18]. To the best of our knowledge, no genome sequences are available for *R. babjevae*. The aim of this study was to obtain high-quality genome assemblies for *R. babjevae* as a prerequisite for the development of genetic tools, and to deepen our understanding of the biology and evolution of *Rhodotorula* species. To achieve this, we used a combination of short and long reads. This has previously been used successfully to generate high quality genome hybrid assemblies in terms of completeness, contiguity, and chromosome reconstruction [3,12,19,20]. We present here the de novo genome assemblies and annotations of two *R. babjevae* species strains, CBS 7808 (type strain) and DBVPG 8058, based on short- and long-read sequencing technologies. We also performed a genome divergence and ploidy analysis of both *R. babjevae* strains.

## 2. Materials and Methods

### 2.1. Yeast Strains

The type strain of *R. babjevae* (CBS 7808) was obtained from the CBS-KNAW collection (Utrecht, The Netherlands). Strain DBVPG 8058 was isolated and identified at the Swedish University of Agricultural Sciences, Uppsala (strain number in the strain collection of the Department of Molecular Sciences is J195) [2] and deposited in the Industrial Yeasts Collection (Perugia, Italy).

### 2.2. DNA Purification

The yeasts were cultivated in 50 mL Yeast–Peptone–Dextrose medium (YPD) until reaching exponential growth phase [21]. Cell wall degradation was performed according to [22] with some modifications. Briefly, the cells were suspended in 1 M sorbitol, 0.1 M sodium citrate, 0.01 M EDTA, and 0.03 M β-mercaptoethanol (SCEM), pH 5.8 after harvesting. Lyticase solution was added to the cell suspensions (100 U/mL) of CBS 7808 and DBVPG 8058, which were then incubated for 9 h or overnight, respectively. After Lyticase digestion, cells were harvested at 1200× *g*, suspended in SCEM buffer, and incubated overnight with Zymolyase (200 U/mL). Genomic DNA extraction from protoplasts was performed using the NucleoBond^®^ CB 20 Kit (Macherey-Nagel, Düren, Germany). DNA concentration, purity, and quality were confirmed through Qubit™ 4 Fluorometer (Thermo Fisher Scientific, Singapore), NanoDrop^®^ ND-1000 Spectrophotometer (Thermo Fisher Scientific, Waltham, MA, USA), and agarose gel electrophoresis, respectively.

### 2.3. Library Preparation and Sequencing

The extracted DNA samples were sequenced using MinION (Oxford Nanopore Technologies, Oxford, UK) and the Illumina sequencing platform. Nanopore DNA libraries were prepared according to [23]. Briefly, 31.5 µL of AMPure magnetic beads were added to 5 µg of DNA for a “pre-cleaning” step. Library preparation was then performed according to a modified protocol [23] using a Ligation Sequencing Kit (SQK-LSK109, Oxford Nanopore Technologies, Oxford, UK). Each DNA library was loaded onto a FLO-MIN106 flow cell mounted on a MINION device (Oxford Nanopore Technologies). MinKNOW software (version 19.06.8) was used for sequencing as described by [23]. The base calling was run using Guppy version 3.2.4-1—195590e and model HAC-mod (modified base sensitive high accuracy model).

From the 6,665,174 long reads recovered from the CBS 7808 DNA library, the mean read length was 2789.7 bases and the read length N50 5553 bases yielding a total of 18,593 Mbp. For DBVPG 8058, 2,953,255 long reads were retrieved containing a total of 15,702 Mbp. The mean read length was 5317 bases and the read length N50 7411 bases. Aliquots of the extracted DNA from both *R. babjevae* strains were also subjected to short-read paired-end sequencing using the Illumina Novaseq platform (S prime, 2× 150 bp) and the TruSeq PCR free DNA library preparation kit (Illumina Inc., San Diego, CA, USA). 179,163,622 short reads were recovered from CBS 7808 DNA library, corresponding to a total of 27,053 Mbp. For DBVPG 8058, 203,873,550 short reads were retrieved containing a total of 30,784 Mbp.

### 2.4. Genome Assembly and Annotation

Genome assembly and annotation was performed using a custom pipeline described elsewhere [3], applying the program versions listed in Appendix A. To further improve the annotation of transcripts and exon–intron boundaries, we additionally mapped RNA-Seq data from the closely related *R. toruloides* CBS 14 (PRJEB40807) to the *R. babjevae* genomes as previously described [3]. We used nQuire (v0.0) based on minimap2 short-read mappings (v2.17; no secondary alignments option) and the KmerCountExact script from the BBMap package (https://sourceforge.net/projects/bbmap/; accessed on 25 November 2021) (v38.86) to estimate the ploidy level of the *R. babje**vae* strains [24,25]. To compare these methods and our ploidy results of the two *R. babjevae* strains with already published results, we also performed nQuire and KmerCountExact on Illumina sequencing data from *Rhodotorula mucilaginosa* JGTA-S1, accession number SRR5821556 [12].

The reconstruction of lipid metabolic pathway maps was performed using KEGG Mapper version 4.3. The KEGG Orthology (KO) identifiers were affiliated to the annotated transcripts of *R. babjevae* CBS 7808 and *R. babjevae* DBVPG 8058 using KofamKOALA [26] with an e-value cut-off of 0.01.

### 2.5. Genome Divergence Analysis

Synteny relationship analysis between *R. babjevae* CBS 7808 and *R. babjevae* DBVPG 8058 was performed using NUCmer (MUMmer, version 3.23). The maximum gap between adjacent matches in a cluster were set to 500 and the minimum cluster length to 100. Visualization of NUCmer alignments and other genomic features was performed with Circa (http://omgenomics.com/circa; accessed on 23 June 2021).

The level of sequence divergence between both *R. babjevae* strains as well as with other closely related *Rhodotorula* species, including *R. glutinis* ZHK (JAAGPT010000000.1), *R. graminis* WP1 (JTAO00000000.1) and *R. toruloides* strains CBS 14 (PRJEB40807), CGMCC 2.1609 (LKER00000000.1), VN1 (SJTE00000000.1) and NBRC 0880 (LCTV00000000.2), was evaluated using the alignment-free distance measure *K*_r_ [27]. We calculated Average Nucleotide Identity (ANI) values using the web-based calculator available at Kostas Lab [28]. DNA–DNA homology (DDH) was estimated with the Genome-to-Genome Distance Calculator (GGDC) 2.1 (http://ggdc.dsmz.de/distcalc2.php; accessed on 26 June 2021) using the program GBDP2_MUMMER [29].

Whole genome alignments of *R. babjevae* strains were performed using LASTZ (version 7.0.2) implemented in Geneious prime, version 2021.0.1 (Biomatters Ltd., Auckland, New Zealand) [30]. Nucleotide alignment and phylogenetic tree construction using MAFFT v7.450 [31] and PhyML 3.3.20180621 [32] with 100 bootstraps, respectively, were performed on the Geneious prime platform.

Whole genome comparison and identification of orthologous gene clusters and paralogous genes were performed on the web-based OrthoVenn2 platform (https://orthovenn2.bioinfotoolkits.net; accessed on 20 October 2021) using a threshold e-value of 1 × 10^−15^ and an inflation of 1.5 [33]. To identify duplicated genes (paralogs) with high sequence identity, an all-against-all sequence identity search was performed on the NCBI Genome Workbench version 3.7.0 [34] using BLASTp (BLOSUM62 matrix) with a cut-off e-value of 1 × 10^−15^. The output file was screened for protein sequences with at least 70% coverage and 70% sequence identity.

## 3. Results and Discussion

### 3.1. Genome Assembly, Ploidy Estimation, and Gene Annotation of R. babjevae Strains

The genome of both *R. babjevae* strains was assembled by a combined approach of long- and short-read sequencing with a coverage depth of about 2000 X. A summary of the genomic data is presented in Table 1. The CBS 7808 draft genome has an overall size of 21,862,387 bp and a GC content of 68.23%. Repetitive sequences make up 5.93% of the total length of the genome, of which 4.98% are single repeats and 0.96% are regions of low complexity. The draft genome of DBVPG 8058 has a total size of 21,522,072 bp and a GC content of 68.24%. The approach identified 6.73% as repetitive sequences, including 5.65% as single repeats and 1.09% as regions of low complexity. The similarity of genome features, such as genome size, GC content, and percentage of repetitive regions, confirms that they are closely related species. The genome size is comparable to that of other *Rhodotorula* species, but the GC content is slightly higher [3,11,12,13,15,18] (Table 1).

Sequence assembly resulted for *R. babjevae* CBS 7808 in 24 contigs and three scaffolds with a length N50 of 1,067,634 bp (Figure 1a, Appendix A). A telomeric region was predicted at one of the termini for 13 contigs and scaffolds larger than 250,000 bp. The draft genome of strain DBVPG 8058 consists of 33 contigs and one scaffold with a length N50 of 789,767 bp (Figure 1b, Appendix A). From the contigs and scaffolds with sizes larger than 250,000 bp in DBVPG 8058 genome assembly, two have telomere sequences at both termini and 15 at one terminus each. The low numbers of contigs and scaffolds in the genome assemblies from both *R. babjevae* strains indicate high accuracy, contiguity, and completeness. Two putative circular sequences were identified in each strain. Among them, contig_2 in CBS 7808 and contig_79 in DBVPG 8058 contained the mitochondrial genes. Both mitochondrial genomes are similar in size with 30.876 bp and 28.432 bp, respectively, and have a GC content of 38.9% (Appendix A).

To estimate the ploidy in *R. babjevae* strains, we used nQuire. nQuire quantifies the distribution of the base frequencies at variable sites, and thus differentiates between different degrees of ploidy [24]. In both strains, the alleles occurred at frequencies of about 25% and 75%, indicating that both *R. babjevae* strains are tetraploid (Figure 2). Furthermore, we also used a k-mer counting approach to estimate ploidy. Using a k-mer length of 31, as recently shown by Sen et al. [12] for *R. mucilaginosa* JGTA-S1, only one peak appears in the plots. However, when the k-mer length is reduced to 17, as recently shown by Zou et al. [35], two distinct peaks appear for both *R. mucilaginosa* JGTA-S1 and the *R. babjevae* strains (Figure 2). The first and larger peak indicates tetraploidy while the second smaller peak indicates diploidy.

The ploidy level of *R. babjevae* strains has not been studied so far. The genomes of the closely related strains *R. toruloides* NP11 and *R. mucilaginosa* JGTA-S1 are considered to be haploid [12,15]. However, our analyses indicate that both *R. mucilaginosa* JGTA-S1 and *R. babjevae* CBS 7808 and DBVPG 8058 may be tetraploid. Tetraploidy has previously been widely recognized in yeast [36,37,38,39]. Knowing the ploidy level is of great importance for genetic engineering and for the development of efficient gene manipulation protocols.

A total of 7591 protein-coding genes and 7607 associated transcripts were annotated in the CBS 7808 genome using MetaEuk (Table 1). The average number of estimated exons per gene is 3.97 (Table 1). The genome of DBVPG 8058 has 7481 protein-coding genes, 7516 associated transcripts and 3.93 estimated exons per gene (Table 1). The proportion of split genes in both genomes is correspondingly high, amounting to 6390 and 6305 for CBS 7808 and DBVPG 8058, respectively. This is consistent with previous findings for *Rhodotorula* spp. [3,12,15]. The distribution of exon counts in the genomes of *R. babjevae* strains CBS 7808 and DBVPG 8058 is shown in Appendix A. 315 and 309 open reading frames (ORF) complementary to annotated genes were predicted in CBS 7808 and DBVPG 8058, respectively. The presence of antisense transcripts has previously been reported for the related species *R. toruloides* [3]. In yeast, the level of antisense transcription has been anti-correlated to sense mRNA, indicating antisense-dependent gene regulation through transcription interference under certain growth conditions [40,41]. Appendix A show the assignment of genes to the Gene Ontology (GO) categories’ biological processes, cellular components, and molecular functions, of which the top 10 are summarized in Figure 3a,b. A total of 2691 and 2660 CDS from CBS 7808 and DBVPG 8058, respectively, could be assigned KO numbers (Figure 3c). The biosynthesis of saturated and unsaturated fatty acids, glycerolipid metabolism, terpenoid backbone biosynthesis, carbon metabolism, and fatty acid metabolism are depicted in detail in Appendix A. Some examples of annotated genes that encode crucial enzymes for lipid and carotenoid metabolism are *CDC19*, *MAE1*, *MAE2*, *ACL1*, *ACL2*, *ACC1*, *FAS1*, *FAS2*, *OLE1*, *ACAD10*, *ACAD11*, *IBR3*, *D6C81_05617*, *POT1*, *LRO1*, *HMG1*, *HCS1*, *ERG8*, *crtYB*, *crtI,* and *BTS1* (Appendix A). A difference in this respect is the absence of *ACL2*, and the presence of *ACAD10* in DBVPG 8058.

Benchmarking of universal single-copy orthologs (BUSCOs, using fungi_odb9) identified that 95.5% and 96.9% of the assessed genes in CBS 7808 and DBVPG 8058, respectively, were complete and single-copy (Appendix A). This supports the high quality of the draft genome assemblies reported here. Furthermore, 0.7% and 0.3% of the assessed genes in CBS 7808 and DBVPG 8058, respectively, were fragmented and the rest were missing (Appendix A). A small percentage of BUSCO genes might still be undetectable due to sequence regions with low coverage, repetitive elements, or assembly problems that cannot be solved even with the hybrid approach and would require additional sequencing and manual analysis. In addition, when a BUSCO gene was missing, there were either no significant matches or the BUSCO matches were below the range of values for the selected BUSCO profile. Finally, some marker genes that are part of the BUSCO “fungi” profile that we used as reference may not be part of the two *R. babjevae* strains.

### 3.2. Chromosome Organization

The *R. babjevae* genome assemblies were aligned for comparison using NUCmer. Out of a total of 27 contigs and scaffolds in CBS 7808, 24 matched 30 of the 34 assembled sequences in DBVPG 8058 (Figure 4). In general, the number of undisturbed segments is high. However, there are also major chromosomal rearrangements (Figure 4). LASTZ alignments of each contig from one *R. babjevae* strain with the whole genome of the other strain confirmed the results of the synteny analysis (Appendix A, Appendix A). Based on these alignments we deduce that *R. babjevae* has a maximum of 21 chromosomes with sizes ranging from 0.4 to 2.4 Mbp (Table 2, Figure 4). The molecular karyotype of several *Saccharomyces* yeast strains has been identified as 16 [42]. Karyotyping studies in *Rhodotorula* species have identified at least 10 chromosomes in isolates of *R. mucilaginosa* and 11 in *R. toruloides* while Martín-Hernández et al. proposed that *R. toruloides* CBS 14 has at least 18 chromosomes [3,43,44]. The pairwise identity between chromosomes ranges from 82% to 87%. The mitochondrial genomes have 97% pairwise identity (Appendix A). Four of the putative chromosomes are affected by large translocation events. This affects chromosomes 3 and 6, and chromosomes 9 and 14 (Table 2). Minor inversions were noticed in other chromosomes (Appendix A). Each *R. babjevae* strain contains two contigs that are strain-specific (Appendix A). These are small linear contigs with higher read depths than the chromosomes, except for circular contig_26 in CBS 7808, which has a lower read depth than the chromosomes. These variations in read depth may indicate relaxed replication regulation. The linear DNA sequence from CBS 7808 contig_46 has two annotated genes, one of which encodes the Retrovirus-related Pol polyprotein from transposon 17.6. DNA plasmids have previously been found in filamentous fungi, including the close relative *R. toruloides*, with sizes ranging from 2.5 to 11 kb and typically encoding enzymes involved in plasmid replication [3,45,46]. This might indicate the presence of strain-specific extrachromosomal endogenous DNA.

### 3.3. Genome Divergence Analysis

The genomes of the *R. babjevae* strains were compared to each other and to genomes of closely related *Rhodotorula* species in terms of DDH, ANI and *K*_r_ for tracing genome divergence (Figure 5, Appendix A). The *R. babjevae* strains share 44.20% DDH estimates, 84.48% ANI and *K*_r_ values of 0.09. In general, the genetic divergence between *R. babjevae* strains was comparable to the divergence with *R. graminis* and *R. glutinis*, but higher than expected for strains of the same yeast species [47]. For instance, the divergence between strains of *R. toruloides* was much lower than that of the two *R. babjevae* strains (Appendix A).

Moreover, the protein-coding sequences of the *R. babjevae* strains and their closest relatives *R. graminis* and *R. glutinis* were analyzed using OrthoVenn2 web platform to identify and compare orthologous gene clusters. The *R. babjevae* species share 6598 out of a total of 7223 orthologous clusters produced by OrthoVenn2, including both single-copy gene clusters and overlapping gene clusters such as paralogs (Figure 6). Of the shared clusters, 5933 are common within the three *Rhodotorula* species assessed, representing putative shared orthologous proteins that evolved from common ancestral genes. In addition, CBS 7808 has 389 single genes and one cluster that had no orthologs in the other genomes, while strain DBVPG 8058 has 355 single genes. These unique genes could account for the specific functional capabilities of the *R. babjevae* strains as a result of gene loss or gain events. Of the 79 orthologous clusters shared only between *R. babjevae* strains, some of the assigned GO terms are: Positive regulation of the unsaturated fatty acid biosynthetic process by positive regulation of transcription from RNA polymerase II promoter (GO:0036083), protein O-linked glycosylation (GO:0006493), glucan catabolic process (GO:0009251), cellular calcium ion homeostasis (GO:0006874), sulfate assimilation (GO:0000103), and carbohydrate transport (GO:0008643). The two *R. babjevae* strains show a high genome pairwise similarity and a high number of shared orthologous clusters, though not as high as for *R. graminis* and *R. glutinis* (Figure 6). In general, *R. babjevae*, *R. glutinis,* and *R. graminis* are very closely related species with a short evolutionary distance between them as compared to other species in the genus (i.e., *R. toruloides*). The strains CBS 7808 and DBVPG 8058 have high interstrain variability and a greater evolutionary distance to *R. graminis* than to *R. glutinis*.

A total of 59 and 30 paralogous gene clusters were identified in CBS 7808 and DBVPG 8058, respectively, using OrthoVenn2 (Appendix A). Applying a cut-off value of 70% sequence coverage to them, we identified 29 and 19 duplicated genes, respectively, that potentially have not diverged in function. On the other hand, an all-against-all protein sequence similarity search was performed in each of the two strains using BLASTp with an e-value of 1 × 10^−15^, 70% coverage, and 70% sequence identity. This resulted in a total of 34 and 21 duplicated sequences in CBS 7808 and DBVPG 8058, respectively, and a total of 41 and 29 duplicated sequences with 70% sequence coverage, respectively, that were identified by any of the tools (Figure 1, Appendix A). The higher accumulation of duplicated genes in CBS 7808 could be related to a higher number of gene duplication events due to faster evolution of the strain. The majority of these duplications lies adjacent to each other or in close proximity. Tandem duplications have been suggested as a mechanism of adaptative evolution to changing environments [48]. They may have arisen through homologous recombination between sequences on sister chromatids or homologous chromosomes [48]. It has been reported that considerable redundancy of duplicate gene pairs persists even after 100 million years of evolution in *Saccharomyces cerevisiae* [49]. Some of the predicted functions of the genes, which are duplicated only in CBS 7808, are Uncharacterized protein C17G8.02 (NAD biosynthesis), Mannose-6-phosphate isomerase and Phosphoenolpyruvate carboxykinase (ATP) (carbon metabolism), Acetyl-CoA carboxylase (fatty acid metabolism), Alpha-ketoglutarate-dependent sulfonate dioxygenase, Sulfite reductase [NADPH] hemoprotein beta-component and Sulfite reductase [NADPH] subunit beta (Sulfur metabolism), and Probable quinate permease (import of quinic acid as a carbon source). Some of the duplicated genes involved in metabolic processes identified only in DBVPG 8058 are mitochondrial Aspartate aminotransferase (intracellular NAD(H) redox balance) and Leucine-rich repeat extensin-like protein 3 (AtLRX3, cell morphogenesis). In both strains, the most frequently duplicated gene is *SRRM2*, which codes for the Ser/Arg repetitive matrix protein 2 and is involved in mRNA splicing. Cwc21p is encoded by *CWC21*, an ortholog of human *SRRM2* in *S. cerevisiae*. It has been proposed that it resides in the catalytic center of the spliceosome and possibly fulfills its role in response to changing cellular environmental conditions [50]. The predicted function Ser/Arg repetitive matrix protein 2 was annotated in 1055 genes in CBS 7808 and 1068 in DBVPG 8058. Alternative splicing is an essential driver of proteomic diversity and may potentially provide a high level of evolutionary plasticity.

The type strain CBS7808 of *R. babjevae* investigated here, was first isolated from herbaceous plants in Moscow, Russia [51]. *R. babjevae* DBVPG 8058 was isolated from wild apples in Uppland locality, Sweden. The phylogenetic placement of DBVPG 8058 to the *R. babjevae* species was performed by the Industrial Yeasts Collection DBVPG by aligning 5.8S-ITS rDNA and D1/D2 26S rDNA regions in a similar manner as illustrated in Figure 5. However, the genome divergence values (DDH, ANI and *K*_r_) proved to be more sensitive for delineating *Rhodotorula* species. Phylogenetic placement based on the standard rDNA regions may not be sufficient to understand yeast diversity and species delineation, as shown before [47,52,53]. These *R. babjevae* strains showed different behavior during enzymatic cell wall degradation for DNA purification both in this study and in another study where xylose medium was used [54]. Highly dynamic genome structures have already been found in closely related yeast species [20,55,56,57,58,59]. A dynamic genome structure of *R. babjevae* could enhance the physiological capabilities and thus the species’ environmental adaptability. [60,61,62,63]. However, their genetic divergence suggests that they may belong to different species. A genome comparison study using whole genome sequences from different strains of closely related *Rhodotorula* species would allow gaining a deeper knowledge about their genome structure and evolution, as well as identifying new species.

Taxonomic classification using Sourmash [64] and the GenBank reference (https://osf.io/4f8n3; accessed on 17 March 2022) assign to both genome assemblies: Eukaryota superkingdom, Basidiomycota phylum, Microbotryomycetes class, Sporidiobolales order, Sporidiobolaceae family, *Rhodotorula* genus, and *Rhodotorula graminis* species. The taxonomic classification might indicate that *R. graminis* was the closest relative of *R. babjevae* with available genomic data. Previous studies have shown a close evolutionary relationship between *R. babjevae* and *R. graminis*, which was also demonstrated here [13,65].

## 4. Conclusions

The hybrid sequencing approach resulted in high-resolution genomes of *R. babjevae* DBVPG 8058 and CBS 7808^T^. Both strains are tetraploid and have a maximum of 21 chromosomes. Some of the chromosomes show large-scale translocation events. Moreover, we demonstrated a high genome divergence between the *R. babjevae* strains, as high as the divergence to other closely related *Rhodotorula* species. This indicates that the two strains do not belong to the same species.

## Figures and Tables

**Figure 1 jof-08-00323-f001:**
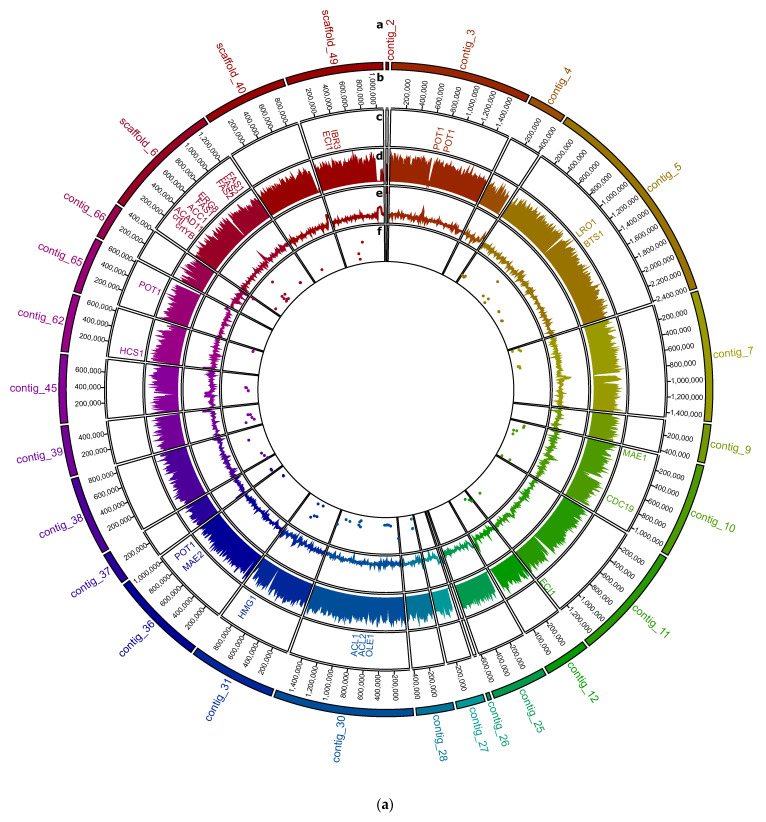
Overview of the genome assemblies of *Rhodotorula babjevae* strains: (**a**) CBS 7808, (**b**) DBVPG 8058. The concentric circles show from outside to inside: the contig name (a) and sizes (b), distribution of lipid and carotenoid metabolism related genes (c), and in non-overlapping 10 kb windows, the gene density (d), the deviation from the average GC content (e), and the density of duplicated genes with 70% sequence coverage (f).

**Figure 2 jof-08-00323-f002:**
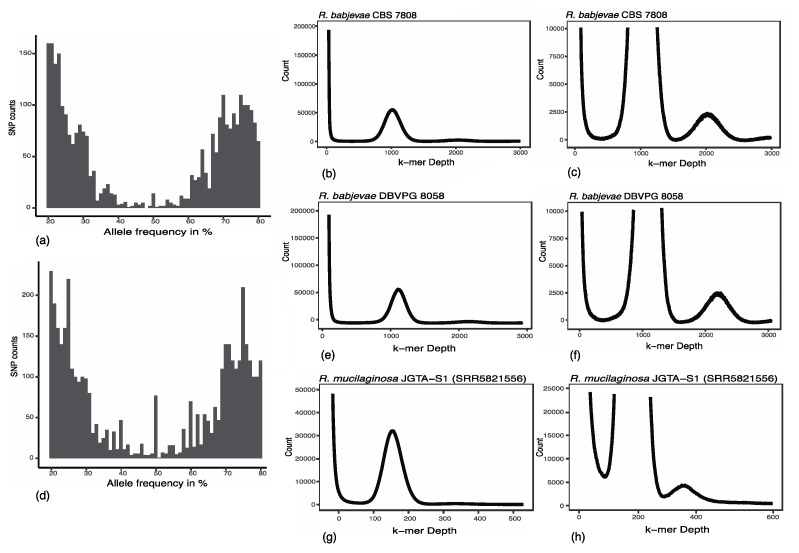
Ploidy estimation of *Rhodotorula babjevae* and *Rhodotorula mucilaginosa*. *R. babjevae* CBS 7808: (**a**) Allele frequency values of single nucleotide polymorphisms (SNP) obtained through nQuire calculations using minimap2, (**b**) Distribution of 17-kmer frequencies using KmerCountExact from the BBmap package, (**c**) Zoomed in peaks of the 17-kmer frequency histogram. *R. babjevae* DBVPG 8058: (**d**) Allele frequency values of SNP (**e**) Distribution of 17-kmer frequencies, (**f**) Zoomed in peaks of the 17-kmer frequency histogram. *R. mucilaginosa* JGTA-S1: (**g**) Distribution of 17-kmer frequencies, (**h**) Zoomed in peaks of the 17-kmer frequency histogram. The reproduction of *R. mucilaginosa* JGTA-S1 ploidy estimation was performed using the Illumina data SRR5821556.

**Figure 3 jof-08-00323-f003:**
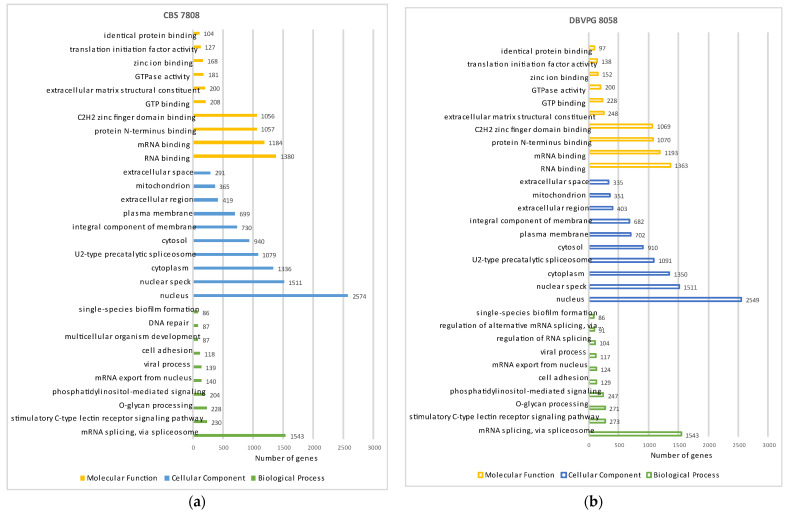
Assigned numbers of genes to the top 10 of the GO categories: biological processes, cellular components, and molecular functions in CBS 7808 (**a**) and DBVPG 8058 (**b**). (**c**) Assigned number of genes to the five KEGG top categories: metabolism, genetic information processing, environmental information processing, cellular processes, and organismal systems. KEGG (Kyoto Encyclopedia of Genes and Genomes), GO (Gene Ontology).

**Figure 4 jof-08-00323-f004:**
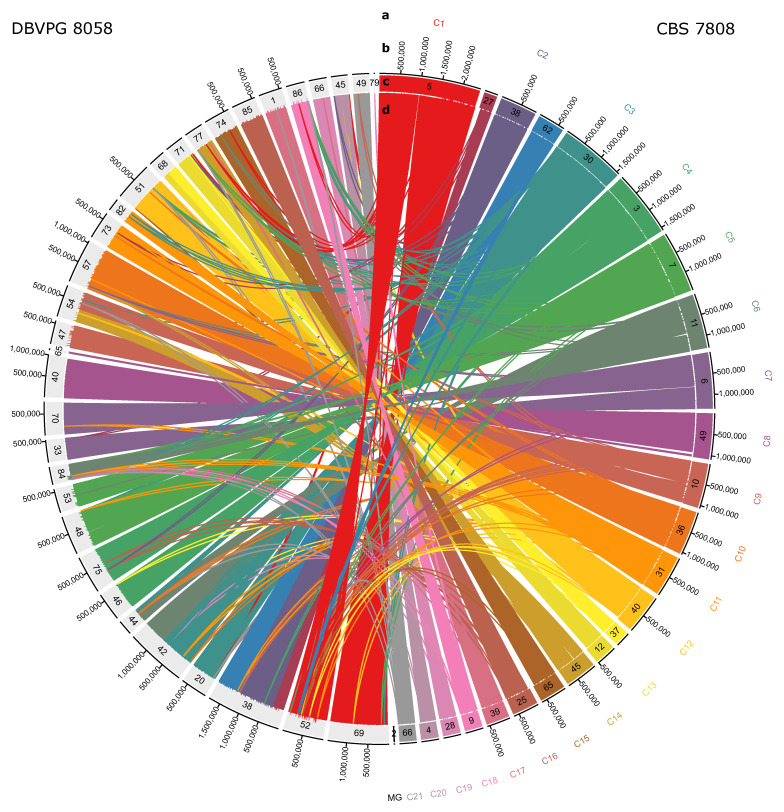
Genome alignment of *Rhodotorula babjevae* strains CBS 7808 and DBVPG 8058. Maximal unique matches between CBS 7808 and DBVPG 8058 were obtained using NUCmer 3.0 and visualized with Circa. The concentric circles show from outside to inside: putative chromosome names or mitochondrial genome, MG, in the reference strain CBS 7808 (a); contig and scaffolds’ sizes (b); and names (c). Ribbons are showing the unique and repetitive alignments using CBS 7808 contigs and scaffolds as the reference (d). Contigs from DBVPG 8058 are colored gray.

**Figure 5 jof-08-00323-f005:**
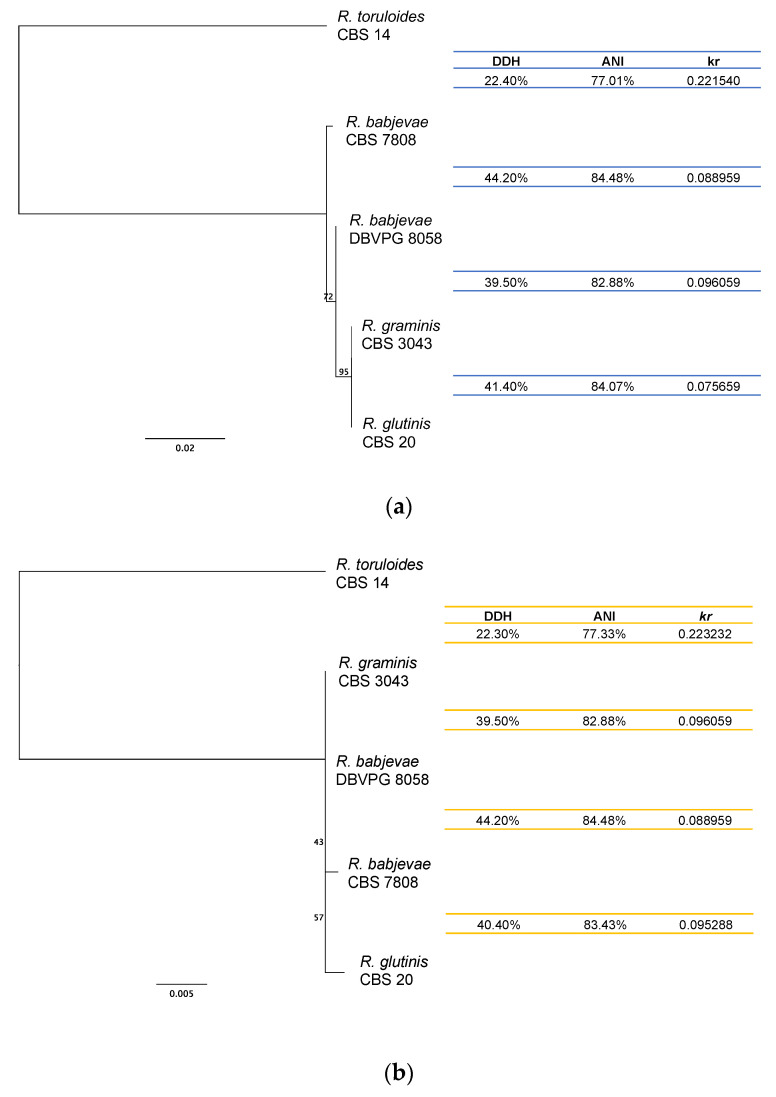
Phylogenetic relationship of *Rhodotorula babjevae* strains and their placement within the *Rhodotorula* genus. The phylogenetic tree was built based on: (**a**) ITS; and (**b**) D1/D2 LSU of rRNA gene sequences. It was inferred using PhyML with 100 bootstraps on Geneious prime version 2021.0.1. *Rhodotorula toruloides* was selected as outgroup. Similarities between whole genome sequences of the corresponding strains are presented in terms of the alignment-free distance measure *kr*, Average Nucleotide Identity (ANI), and DNA–DNA homology (DDH). *Rhodotorula graminis* WP1 and *R. glutinis* ZHK genome sequences were used for the calculations instead of *R. graminis* CBS 3043 and *R. glutinis* CBS 20, respectively.

**Figure 6 jof-08-00323-f006:**
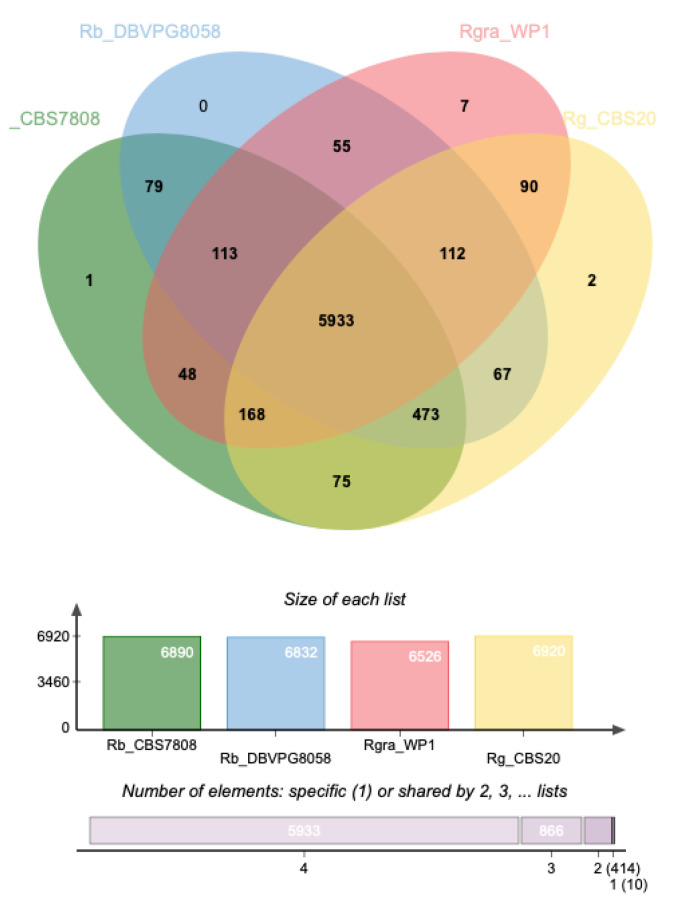
Distribution of shared orthologous clusters between *Rhodotorula babjevae* strains CBS 7808 and DBVPG 8058, *R. graminis* WP1 and *R. glutinis* CBS 20. The Venn diagram was generated using OrthoVenn2.

**Table 1 jof-08-00323-t001:** Genomic data from *Rhodotorula* species.

Reference	This Study	This Study	[11]	[13]	[3]	[15]
Strain number	*R. babjevae* CBS 7808	*R. babjevae* DBVPG 8058	*R. graminis* WP1	*R. glutinis* ZHK	*R. toruloides* CBS 14	*R. toruloides* NP11
Genome size (Mbp)	21.9	21.5	21.0	21.8	20.5	20.2
Coverage	2058	2122	8.6	470	1514	96
GC content (%)	68.23	68.24	67.76	67.8	61.83	62.05
Bases masked (%)	5.93	6.73	6.5	NA	2.01	2.53
No. Scaffolds	3	1	26	30	3	34
No. Contigs	24	33	325	NA	23	NA
Protein-coding genes	7591	7481	7283 _a_	6774 _a_	9464	8171
Avg. no. exons per gene	4.0	3.9	6.2	NA	5.9	NA
Sequencing platform	Nanopore and Illumina	Nanopore and Illumina	Sanger	PacBio and Illumina	Nanopore and Illumina	Illumina and Sanger

NA—not available; _a_—refers to predicted genes.

**Table 2 jof-08-00323-t002:** Putative chromosomes in *Rhodotorula babjevae* deduced from whole genome LASTZ alignments (Appendix A).

*R. bajevae* CBS 7808	*R. bajevae* DBVPG 8058	Genetic Structure	GC Content	Comments	Size (Mbp)
Contig_5 (2,415,752 bp)	Contig_69 (1,447,990 bp)	Putative chromosome 1	67–69%	Appendix A	2.4
	Scaffold_52 (977,625 bp)				
Contig_27 (320,063 bp)	Contig_38 (1,780,658 bp)	Putative chromosome 2	67–69%	Appendix A	1.8
Contig_38 (881,966 bp)					
Contig_62 (644,441 bp)					
Contig_30 (1,569,459 bp)	Contig_20 (637,402 bp)	Putative chromosome 3	67–69%	Appendix A	1.6
	Contig_42 (1,446,680 bp)			Large translocation event between Chr. 3 and Chr.6	
	Contig_44 (357,974 bp)				
Contig_3 (1,574,520 bp)	Contig_46 (670,828 bp)	Putative chromosome 4	67–69%	Appendix A	1.6
	Contig_75 (900,917 bp)				
Contig_7 (1,460,653 bp)	Contig_48 (931,129 bp)	Putative chromosome 5	67–69%	Appendix A	1.5
	Contig_53 (571,073 bp)				
Contig_11 (1,300,441 bp)	Contig_42 (1,446,680 bp)	Putative chromosome 6	67–69%	Appendix A	1.3
	Contig_44 (357,974 bp)			Large translocation event between Chr. 3 and Chr.6	
	Contig_84 (425,340 bp)				
Scaffold_6 (1,337,997 bp)	Contig_33 (529,001 bp)	Putative chromosome 7	67–69%	Appendix A	1.3
	Contig_70 (789,767 bp)				
Scaffold_49 (1,089,446 bp)	Contig_40 (1,004,683 bp)	Putative chromosome 8	67–69%	Appendix A	1.1
	Contig_65 (41,334 bp)				
Contig_10 (1,067,634 bp)	Contig_47 (557,103 bp)	Putative chromosome 9	67–69%	Large translocation event between Chr. 9 and Chr.14	1.1
	Contig_54 (766,724 bp)			Appendix A	
Contig_36 (1,056,323 bp)	Contig_57 (1,049,892 bp)	Putative chromosome 10	67–69%	Appendix A	1.1
Contig_31 (979,228 bp)	Contig_73 (659,761 bp)	Putative chromosome 11	67–69%	Appendix A	1.0
	Contig_82 (299,180 bp)				
Scaffold_40 (948,604 bp)	Contig_51 (924,743 bp)	Putative chromosome 12	67–69%	Appendix A	0.9
Contig 37 (362,520 bp)	Contig_68 (408,627 bp)	Putative chromosome 13	67–69%	Appendix A	0.9
Contig_12 (511,897 bp)	Contig_71 (449,691 bp)				
Contig_45 (762,860 bp)	Contig_54 (766,724 bp)	Putative chromosome 14	67–69%	Appendix A Large translocation event between Chr. 9 and Chr.14	0.8
	Contig_77 (446,828 bp)				
Contig_65 (630,535 bp)	Contig_74 (614,034 bp)	Putative chromosome 15	67–69%	Appendix A	0.6
Contig_25 (627,118 bp)	Contig_85 (573,802 bp)	Putative chromosome 16	67–69%	Appendix A	0.6
Contig_39 (564,129 bp)	Contig_1 (565,532 bp)	Putative chromosome 17	67–69%	Appendix A	0.6
Contig_9 (429,397 bp)	Contig_86 (443,617 bp)	Putative chromosome 18	67–69%	Appendix A	0.4
Contig_28 (422,133 bp)	Contig_66 (419,035 bp)	Putative chromosome 19	67–69%	Appendix A	0.4
Contig_4 (418,972 bp)	Contig_45 (394,205 bp)	Putative chromosome 20	67–69%	Appendix A	0.4
Contig_66 (406,102 bp)	Contig_49 (396,114 bp)	Putative chromosome 21	67–69%	Appendix A	0.4

Chr., chromosome.

## Data Availability

This project has been deposited at ENA under the accession PRJEB48745.

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
