# Peer review of "Near Chromosome-Level Genome Assembly and Annotation of Rhodotorula babjevae Strains Reveals High Intraspecific Divergence"

_jof, 2022, doi:10.3390/jof8040323_

Round 1

Reviewer 1 Report

The manuscript by Martín-Hernández et al aims to provide a well annotated genome of R. babjevae. For this, the authors performed nanopore and Illumina sequencing, and provide a series of analyses of the genome structure and some functional analyses. Overall, is a nice paper, however, I believe that many sections of the manuscript can be improved to further provide the reader with elements and figures to understand the genome of R. babjevae. I have several major comments that I would suggest the authors to address

  1. The introduction does not mention anything about other Rhodotorula genomes. I suggest including this information from the beginning since is considered in results and discussion.
  2. Ploidy should be corroborated using FACS analysis
  3. Figure 1 can be improved. Gene names are on top of each other, and the legend itself is not informative enough to understand the figure
  4. Why do the authors talk about antisense transcripts in the results first section? I would suggest removing this sentence
  5. Discussion should be improved throughout the whole manuscript. Not many references are provided, and the genomes don't seem to be compared to other genera or genomes which are also relevant for yeast genetics
  6. I suggest moving Figure 2 to Supplementary Material
  7. BUSCO completeness values are below 97%. Can the authors discuss this more?
  8. I suggest showing Structural variants in a single synteny plot as a main figure
  9. Tables 2 and 3 should go to supplementary methods
  10. Can the authors provide a putative chromosome karyotype? Also would be nice to support this with CHEF gels
  11. I think if whole genomes are available, phylogenetic trees in figure 4 should use whole genomes (using orthologues) rather than only ITS and D1/D2
  12. If the authors believe that these are different species, then a species delimitation analysis should be performed.

Author Response

We thank the referees for the time they took to review our manuscript and for their valuable comments and suggestions. We regret our late comment. Due to the Christmas and New Years holiday, the bioinformatics analyzes that were necessary to specifically address one point made by the referees have been delayed. We have considered and addressed all points as listed below.

On behalf of all authors.

Sincerely,

Bettina Müller

Point-by-point answer to the referees’ comments and suggestions:

Reviewer #1:

The manuscript by Martín-Hernández et al aims to provide a well annotated genome of R. babjevae. For this, the authors performed nanopore and Illumina sequencing, and provide a series of analyses of the genome structure and some functional analyses. Overall, is a nice paper, however, I believe that many sections of the manuscript can be improved to further provide the reader with elements and figures to understand the genome of R. babjevae. I have several major comments that I would suggest the authors to address.

  1. The introduction does not mention anything about other Rhodotorula genomes. I suggest including this information from the beginning since is considered in results and discussion.

We added information about Rhodotorula genomes to the introduction:

L56-60: To date, several genomes from Rhodotorula sp. have been sequenced including different strains from R. toruloides, R. graminis WP1 and R. glutinis ZHK. Some of them have only been determined using short-read sequencing technologies or lack gene annotation [3,11–18]. To the best of our knowledge, no genome sequences are available for R. babjevae.

  1. Ploidy should be corroborated using FACS analysis

That would indeed be a methodological approach that could further shed light on the ploidy of the strains presented here. Unfortunately, we don't have this technology in-house and would first have to establish a collaboration, which is currently being made even more difficult by the pandemic. Furthermore, we could currently only determine the strains in relation to each other, since the ploidy of the Rhodotorula strains is rather unclear and what to take as “a standard” for comparing DNA content is therefore difficult. However, we are working on a follow-up study involving a variety of Rhodotorula strains to further investigate ploidy, among others. In the study presented here, we used a bioinformatics only approach to investigate the ploidy as further elaborated below (point 13).

  1. Figure 1 can be improved. Gene names are on top of each other, and the legend itself is not informative enough to understand the figure

Figure 1 was modified as suggested.

  1. Why do the authors talk about antisense transcripts in the results first section? I would suggest removing this sentence

We found signals of potentially interesting antisense genes. We did not investigate this further in this study but wanted to mention it because such antisense genes have been observed occasionally, including in the close relative Rhodotorula toruloides. Their function has been linked to sense gene regulation. The text has been changed to emphasize their importance in gene regulation.

L226-229: The presence of anitisense transcripts has previously been reported for the related species R. toruloides [3]. In yeast, the level of antisense transcription has been anti-correlated to sense mRNA, indicating antisense-dependent gene regulation through transcription interference under certain growth conditions [38,39].

  1. Discussion should be improved throughout the whole manuscript. Not many references are provided, and the genomes don't seem to be compared to other genera or genomes which are also relevant for yeast genetics.

L277-L280: The molecular karyotype of several Saccharomyces yeast strains has been identified as 16 [40]. Karyotyping studies in Rhodotorula species have identified at least 10 chromosomes in isolates of R. mucilaginosa and 11 in R. toruloides while Martín-Hernández et al. proposed that R. toruloides CBS 14 had at least 18 chromosomes [3,41,42].

We further improved the discussion with regards to other comments and suggestions, see below.

  1. I suggest moving Figure 2 to Supplementary Material

Figure 2 was moved to Supplementary Material as suggested.

  1. BUSCO completeness values are below 97%. Can the authors discuss this more?

Here, we used the ‘fungi’ BUSCO (Benchmarking universal single-copy orthologs) set to assess the completeness of our assembly with regards to these marker genes. However, we can not be 100% certain that all marker genes that were selected for the BUSCO set are actually part of our target genome. For example, see the Figure at the bottom of this page (https://busco.ezlab.org/) that shows that, although marker BUSCO genes are selected to cover a high universality of the respective taxon, the cutoff is at >90% of the selected species. Accordingly, there might be marker BUSCOs in the “fungi” profile that are present in 90% of the reference species but not in our R. babjevae strains. Thus, we might not be even able to achieve 100% even with a perfect and complete assembly. However, on the other hand, we can not fully exclude that the assemblies might still contain few errors (low coverage sequence regions, repetitive regions, assembly issues, ...) that can lead to missing/incomplete genes, which might account for the ~3% missing BUSCOs. However, in our experience a complete BUSCO detection rate of 97% is already quite high and correlates with a high-quality genome assembly in terms of the used reference BUSCO set. We added an interpretation of these BUSCO values to Results & Discussion to better explain these values:

L241-252: Benchmarking of universal single-copy orthologs (BUSCOs, using fungi_odb9) identified that 95.5% and 96.9% of the assessed genes in CBS 7808 and DBVPG 8058, respectively, were complete and single-copy (Figure S7). This supports the high quality of the draft genome assemblies reported herein. Furthermore, 0.7 % and 0.3% of the assessed genes were fragmented in CBS 7808 and DBVPG 8058, respectively, and the rest were missing (Figure S7).  A small percentage of BUSCO genes might still be undetectable because of sequence regions with low coverage, repetitive elements, or assembly problems that cannot be solved even with the hybrid approach and would require additional sequencing and manual analysis. In addition, if a BUSCO gene was missing, there were either no significant matches at all or the BUSCO matches were below the range of values for the selected BUSCO profile. Finally, some marker genes that are part of the BUSCO "fungi" profile that we used as reference may not be part of our R. babjevae strains.

  1. I suggest showing Structural variants in a single synteny plot as a main figure

We have modified the circa plot in Figure 2 that shows unique and repetitive alignments between the genomes of the two described R. babjevae strains to improve readability. A more detailed view can be found in Figure S9.

  1. Tables 2 and 3 should go to supplementary methods

Table 3 was moved to Supplementary Material as suggested. However, we would like to keep table 2 in the main manuscript.

  1. Can the authors provide a putative chromosome karyotype? Also would be nice to support this with CHEF gels

Thanks for the suggestion, but unfortunately, we don't have the technical opportunity to run CHEF in our lab. Also, for chromosomes of similar size such an analysis would not always give fully conclusive results. We tried to cover this at least via including long-read Nanopore sequencing data.

  1. I think if whole genomes are available, phylogenetic trees in figure 4 should use whole genomes (using orthologues) rather than only ITS and D1/D2

Unfortunately, from the assessed Rhodotorula species we only have four genomes available and one of them is not annotated. Besides, the phylogenetic trees in figure 4 (current figure 3) was built to show i) how the previous assignment of these two strains to the Rhodotorula babjevae species was carried out through sequencing of the 5.8S-ITS rDNA and D1/D2 26S rDNA regions, and that ii) this method is not as sensitive for delineating Rhodotorula species. D1/D2 and/or ITS sequence analyses should be seen as a guideline for the recognition of novel taxa and a first estimation of the closest known species but for identifying the closest relative, the between-species OGRI (overall genome related indices) such as ANI, dDDH and kr distance are more suitable when genome sequences are available (Libkind et al. 2020).

The text was modified to make it more understandable:

L498-501: The phylogenetic placement of DBVPG 8058 to the R. babjevae species was done by the Industrial Yeasts Collection DBVPG through aligning 5.8S-ITS rDNA and D1/D2 26S rDNA regions in a similar way as illustrated in Figure 3.

  1. If the authors believe that these are different species, then a species delimitation analysis should be performed.

Our results indicate that these genomes may represent different species. Nevertheless, this is a hypothesis that comes out from the genome analysis, it is not a believe. Performing a comprehensive species delimitation analysis would require more studies, including analysis of more strains of the potential new species, which are currently not available. This would clearly go beyond the scope of this study, which was the characterization of genomes of strains of R. babjevae.

Reviewer 2 Report

The manuscript “Near chromosome-level genome assembly and annotation of Rhodotorula babjevae strains" reveals high intraspecific divergence” by Martín-Hernández and colleagues reports the genome sequencing and comparison of two Rhodotorula babjevae strains (one of which being the type strain). The comparison revealed some interesting chromosome plasticity, with relevant chromosomal rearrangements. Furthermore, the comparison of the genomic sequences of these two strains with these of other Rhodotorula species highlighted unexpectedly high intra-specific divergence, possibly indicating the inappropriate assignment of the two strains to the same species and suggesting the existence of a new Rhodotorula species.

Despite being of great interest and providing a major improvement on the genomics of Rhodotorula species, the manuscript clarity could be improved as detailed below.

Ploidy: it is interesting to note that the R. babjevae DBVPG 8058 strain, in addition to showing a large portion of SNPs with 1:4 alleles also shows a similar amount of SNPs with 50% allele frequency, differently from the R. babjevae CBS 7808 strain. Despite the authors have not highlighted this point in the text, I think this could provide additional interesting insights on the genome of this strain. First of all, this observation should be evaluated in terms of chromosomal ploidies, e.g. the authors should take advantage of the Illumina sequencing data to estimate the number of copies of each scaffold/contig. Are there any aneuploidies in the DBVPG 8058? In fact, the neat signal around the 50% allele frequency could indicate loss of chromosomal copies. Otherwise, this could indicate a different origin of the strain.

Figure 2 should be largely modified to improve readability. First of all, bigger font size should be used. Secondly, the two images could be merged to facilitate comparison: e.g. two different colors could be used corresponding to the two strains, and a third color could be used to show features shared by both strains. I know these are images automatically generated by KEGG mapper, but I believe modifying/reproducing them would greatly help the reader appreciate the data and differences.

L214: I am not familiar with BUSCO, and I am afraid I do not understand what “single-copy” means here… how was the gene copy number determined? Since the strains were identified as being tetraploid, it is very unlikely that a large number of genes were single-copy…

As for figure 2, also figure 3 could be greatly improved. Scaffold/contigs should be better sorted to reduce the number of crossings of ribbons indicating matches in the alignment. At a glance, it is hard to tell that, as the authors state “a high proportion of chromosomal rearrangements can be spotted” (L222-223). Contrarily, by observing the synteny plots shown in Supplementary Figure 9, chromosomal rearrangements are much more obvious. However, I would invite the authors to check some of the information reported in these figures – and hence the corresponding conclusions… In particular, I am referring to the observation of “forward” and “reverse” alignments. For instance, in panels c, d, and f of Supplementary Figure 9, the “reverse” alignment corresponds to entire contigs of one strain, or to the part of contigs of one strain matching with the indicated contig of the other strain. Authors should consider that the “direction” of the contig, as obtained through sequencing, is arbitrary, and hence consider that it is more likely that that region of the chromosome is actually collinear among the two strains, but the analysis resulted in the complementary reverse sequence with respect to the rest of the chromosome. Or am I missing something? If not, the current display misleads the interpretation of results. Please consider modifying them.

Also, I honestly cannot see from the figures the supposed translocations between Chromosomes 3 and 6 and between 9 and 14…

Another very striking observation is the fact that the genomic difference between the two Rhodotorula babjevae strains exceeds the difference among these strains and the sequence of closely related Rhodotorula species. The authors correctly suggest that the previous assignment of these two strains to the Rhodotorula babjevae species, carried out through sequencing of the 5.8S-ITS rDNA and D1/D2 26S rDNA regions, could be wrong and that these two strains could actually belong to different species (hence suggesting the existence of a previously unknown species!). Considering the relevance of this finding, even if not confirmed in the current study, it should be reported in the abstract.

Minor comments:

Please note that most of the Supplementary Tables are not fully displayed in the file (portrait vs landscape page orientation).

Authors indicate that, among others, the ERG8 gene is missing in the DBVPG 8058 strain… this gene is quite relevant in fungal physiology and morphology, as it is involved in the synthesis of ergosterol, one of the components of the fungal cell wall. Do the strain show atypical phenotypes in this optic?

Reviewer 3 Report

The article describes in detail the methods and results. Everything is quite clear and understandable. I have no further comments. I think that the article should be accepted in its current form.

Author Response

No revision requested.

Reviewer 4 Report

The authors have sequenced and annotated two genomes of two strains of Rhodotorula babjevae. The authors further compared these two genomes with other species in the same genus. I believe that this manuscript provides the fundamental baseline information of the genomes of these two strains.

My major concerns are that I think it is important to present the GO and KEGG data in 1-2 figures to be present in the main text, and I also recommend that the authors should be selective in the conclusion of the supplementary materials. Additionally, I strongly believe that the English of this manuscript needs to be significantly improved. I have listed the following grammatical and editorial suggestions and some minor technical concerns (but I know there are more throughout the entire manuscript) for authors to consider if a revision is requested by the editor.

Title: remove “Near chromosome-level” from the title. Also, it is not clear why the intraspecific divergence is found whereas the manuscript describes the largely similar genomes of these two strains.

Line 17, a sentence cannot be started with an abbreviation of “R.”

Lines 22 and 24, please be consistent either Mbp or Mb.

Lines 27 and 28, please indicate which database(s) are used in the annotation.

Lines 54 and 55, please correct “some of them”

In the last paragraph of the Introduction, the authors should have specified the goals of their study.

Lines 67 and 68, a sentence cannot be started with an abbreviation of “R.” throughout the entire manuscript.

Lines 69 and 70, the authors need to clarify what SLU is and what institute the Dept of Mol Sci belongs to..

Line 72, what is YPD? And in two lines below, what is SCEM?

Lines 95-104, remove all “sequenced” in this paragraph.

Line 110, please correct the use “like also shown before.”

Line 115, from, not form.

Line 117, please find the correct location of this orphaned sentence.

Line 140, remove “too”

Table, the first two rolls should be combined to make one roll; also “a” needs to be a superscript in the table, otherwise, which “a” is the “a” used to indicate the predicted genes?

Lines 166 and 169, 1A and 1B but it is “a” and “b” in Figure 1, which should be corrected to “A” and “B”

Line 176, Tables S2 and S3.

Lines 187 and 188, please rewrite this sentence, it is confusing…

Line 196, analysis but not analyzes.

Lines 198 and 199, please rewrite this sentence, it is confusing…

Line 206, “spp.”

Line 269, Table S7.

Again, there are more in the rest of the manuscript but I stopped here…

Round 2

Reviewer 2 Report

The authors have carefully approached all my concerns and properly addressed them. I believe the study, which was already interesting and relevant, has greatly improved and I am sure it will be well-received by scientists studying Rodothorula and in general yeast genomics and evolution.

Reviewer 4 Report

I appreciate the authors' efforts to improve their manuscript. However, I think the authors may have misunderstood my suggestion of presenting the GO and KEGG annotation results. I will explain here what I meant. Again, I  believe that it is necessary to present the GO and KEGG annotations in the bar graphs, representing the total number of genes annotated into those three categories (any any subcategories) in GO database and the the total number of genes annotated into the metabolic pathways in KEGG database, for the two strains, but not the format as provided currently in the supplementary materials Figures 2-4 and Figures 5-6. As I indicated previously, the inclusion of data in the supplementary materials should be selective. The supplementary figures 2-4 should be presented in two bar graphs for the two strains and the supplementary figures 5 and 6 (KEGG annotation) should be presented in two bar graphs for the two strains as well, both in the main text, not in the supplementary materials. I also believe that the bar graphs are extremely commonly presented for this type of studies. I apologize for not making myself clear and I hope I have explained this clearly now.

Author Response

Many thanks to the reviewer for the clarification. We have presented both the GO and the KEGG classifications as bargraphs and integrated them into the main text.

Round 3

Reviewer 4 Report

Thanks for the inclusion of both GO and KEGG annotations. I have no further questions.